

# HESS Opinions: Agricultural irrigation with effluent – Pharmaceutical residues that we should worried about

Dror Avisar[1], Gefen Ronen-Eliraz[1]

[1] The Water Research Center, Hydrochemistry Research Group, Porter School of Environment and Earth Sciences, Faculty of
Exact Sciences, Tel Aviv University, Tel Aviv 69978, Israel

*Correspondence to*: Dror Avisar (droravi@post.tau.ac.il)

**Abstract.** Policy regarding effluent water and reclamation aims to prevent environmental pollution while proposing an alternative water resource. Water makes up 99–99.9% of raw wastewater. Thus extracting organic and inorganic matter from water is a must. Worldwide, but especially in developed countries, great effort has been made to reuse wastewater, and it is
becoming a reliable alternative source. Israel is the world leader in water reuse, allocating 85% of effluent water for agricultural irrigation. As such, it constitutes a "living laboratory" in which to study the implications of the intensive use of treated wastewater for agricultural irrigation, leading to research and legislation regarding effluent quality and regulation. Effluent produced in Israel is subject to severe regulations and standards and is considered suitable for every use except drinking water. It is mostly allocated for agricultural irrigation with no restrictions. The irrigated lands are close to natural water sources, and
therefore water leaching from the fields infiltrate those sources, becoming part of the water cycle. A group of persistent and toxic nano- and micro-organic contaminants, including pharmaceutical residues, flows to water-treatment plants from hospitals, industry, agriculture and especially the domestic sector. These contaminants' chemical structure, characterized by a couple of aromatic rings and double bonds, makes them especially persistent; they are resistant to conventional biological treatment, used as a secondary treatment. As a result, the effluent that leaves the treatment plants, which is considered to be of
high quality, actually contains pharmaceutical residues. After secondary and tertiary treatment, these persistent chemical residues can still be found in surface water, groundwater and agricultural products. Pharmaceutical residues in effluent allocated for agricultural irrigation are undesirable. Expansion of the monitoring system for those contaminants, improvement of the tertiary treatment, and implementation of advanced technologies for decomposition and removal of pharmaceutical contaminants are thus needed.

**1. Introduction**

Water scarcity is a global problem. In 2017, 29% of the world's population had no access to safely managed water (Ghebreyesus and Lake, 2017; Grojec, 2017). According to the World Health Organization (WHO), in 2012, over 800,000 deaths worldwide were caused by contaminated drinking water, inadequate handwashing facilities and inappropriate sanitation services. The water-scarcity problem has two core aspects: water resource availability and ambient water quality. Both aspects
are directly related to wastewater. Water resources are dwindling due to over-abstraction, pollution, and climate change.


Furthermore, water demand is predicted to increase significantly in the decades to come, to sustain the acceleration in agriculture, industry and energy production (UN-Water, 2017).

Wastewater production is analogous to water demand: it will keep increasing as long as the demand for water does, since most activities that use water produce wastewater in even higher quantities (Fig. 1). Worldwide, 80% of wastewater is released to

5 the environment without adequate treatment (UN-Water, 2015). Untreated wastewater generates chemical, physical and biological pollution, impacting both human health and the environment (UN-Water, 2017). Even when wastewater is collected and treated, its quality is not always sufficient. Wastewater is composed of roughly 99% water and 1% suspended, colloidal and dissolved solids. Ignoring the opportunities offered by improved wastewater management and reuse is therefore unthinkable.

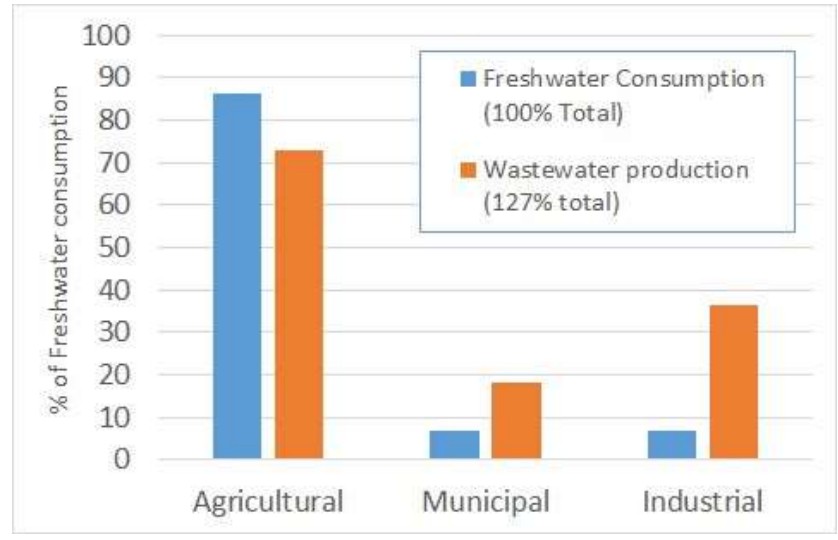

**Figure 1: Water consumption related to wastewater production (Data from UN-Water, 2017).**

## 2. Global water reuse

Sato et al, (2013) analyze the worldwide water reuse accordingly to countrie's level of income. Base on their statistics, in high-income countries, an average 70% of the wastewater is treated. Middle-income countries treat between 28 and 38% of their

15 wastewater volume, whereas in low-income countries, only about 8% of the wastewater is treated (Sato et al., 2013) The quality of the treated wastewater is variable, depending on its source (municipal, industrial or agricultural) and wastewater-treatment plant operation and technology. Despite the expectation that treated wastewater will become the main alternative solution for the water-scarcity problem, the percentage of reuse, in most cases, remains small.



Israel is an example of a country in which reuse treated wastewater and it is actually a leading country in water reuse. Located in a region where water scarcity is a severe problem and natural resources have already been maximally exploited, alternative water resources play a major role in the water sector. Wastewater reuse amounts to 85% of secondary effluent, allocated mainly for agricultural irrigation, whereas in Spain, which is ranked second to Israel, this percentage is only 25. The other alternative

5 water resource is desalinated water and together, these alternatives are expected to expand to 68% of the total water resources in 2040 (Fig. 2) (Bar-Eli, 2017).

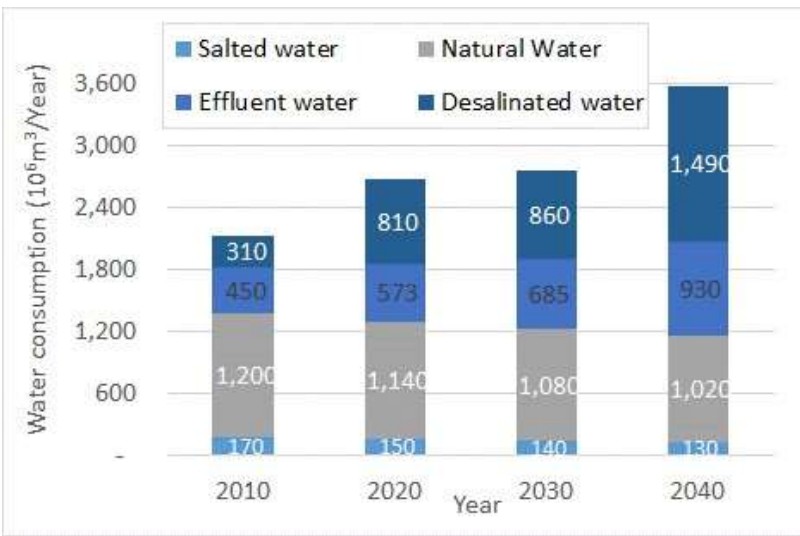

Figure 2: Estimates of water resource consumption up until 2040 in the Israeli water sector.

### 3. The complications of wastewater reuse

10 Wastewater is composed of many different compounds, depending on its origin (municipal, industry or agriculture). In contrast to fresh water, wastewater contains organic and inorganic contaminants, e.g. pharmaceutical or toxic metals, as dissolved and suspended particles (Grossman and Rurman, 2007; WHO, 2006; Yeh et al., 2015). Recycling of treated wastewater can lead to secondary contamination of soil and vegetation, surface water and adjacent groundwater as a result of these additional chemical compounds (WHO, 2006). As a leading country in allocating treated wastewater for agricultural irrigation, Israel is

15 in essence a "living laboratory" that is committed to carefully examining the various issues and implications that are tightly bound to unlimited wastewater reuse (Cohen et al., 2016).

There has been much research over the years regarding the potential damage to soils and vegetation as a result of irrigation with effluent containing inorganic contaminants (Scott et al., 2004; Mamedov et al., 2000; Dvorkin et al., 2012). Investigations of the occurrence, influence and toxicity of common organic contaminants, such as fertilizers and oil compounds, among



others, have also been published. These types of contaminants, which are known for their negative effects on the environment and human health, are already under investigation, including monitoring, analysis and regulatory restrictions. In contrast, groups of resistant organic micropollutants (OMPs), originating from the pharmaceutical and drug industries and which may threaten human health, are not yet subjected to monitoring or regulatory restrictions, in most of the countries around the world.

Therefore, in this paper, we examine the complexity of the implications arising from recycling wastewater which contains compounds that are neither analyzed nor regulated, but should be treated with caution. Answering the following questions will serve to clarify these implications: What other components are present in effluents? At what concentrations? What are their degradation products? How chemically stable, and how toxic are these degradation products? How do these components and degradation products impact the irrigated environment? What are the regulated parameters in water regulation? Do they

provide any parameters in the context of recycled water? Do the regulated parameters indeed define high-quality effluents?

### 3.1 What components are present in high-grade effluents? At what concentrations?

The wastewater flowing into wastewater plants carries the remnants of persistent organic compounds characterized by a couple of aromatic rings and double bonds, which limit their biological decomposition. Drug residues are an example of such persistent compounds. Depending on the specific circumstances, studies have shown that about 30–90% of all medications

consumed are excreted via the urine and feces as the parent compound or its metabolites (Adamczak et al., 2012; Christou et al., 2017).

A known treatment procedure for wastewater includes pretreatment, which is basically filtration and sedimentation, a secondary treatment which is based on biological degradation, and a tertiary treatment which consists of filtration and disinfection. Sometimes the tertiary stage also includes applicable advanced technologies to degrade specific compounds. The

secondary treatment, which is the main stage for removal of dissolved organic compounds, may remove more than 90% of the degraded organic compounds, as measured by biological oxygen demand (BOD) (Grossman and Rurman, 2007). At the end of the treatment procedure, the treated wastewater is considered "high-quality effluent" as it meets the standards for the main tested parameters, including BOD, COD, TSS, Turbidity, $N_{Total}$, $NO_3^-$, $NH_4^+$, Oil and Grease, etc.

Studies have shown that during the biological (secondary) treatment of wastewater, resistant compounds, such as drug residues,

do not degrade easily. The potential content of drug residues in wastewater estimated based on the difference between the drug's consumption and its percentage in the feces and urine. The global consumption of popular medicines, such as carbamazepine (treats epileptic spasm) and diclofenac (a non-steroidal anti-inflammatory), has been estimated in some western regions around the world, having some assumptions for simplifying (Zhang et al., 2008). Those medications and many others have been found and analyzed in wastewater all over the world. It is important to emphasize, however, that their concentration

in wastewater is on the order of only a few nanograms to micrograms per liter (Lamm et al., 2009; Shafrir and Avisar, 2012; Zhang et al., 2008).




### 3.2 Degradation products – Are they chemically stable? Are they toxic?

Despite their chemical stability, some of the drugs might be degraded while flowing with the wastewater (Wang and Wang, 2016). In addition, natural degradation can be caused by solar radiation, wastewater acidity (pH), or integration of additional organic compounds such as humic acid (Gozlan et al., 2010, 2014). Variation in wastewater composition influences

degradation processes differently, although the parent compound will degrade preferentially to different specific degradation products under different conditions. This further complicates the monitoring and analysis of degradation products (Gozlan et al., 2013, 2016).

Amoxicillin (AMX) is the active ingredient in commonly consumed antibiotics worldwide. Surprisingly, it has never been detected in wastewater or effluent. It was found that AMX degrades easily due to opening of its strained β-lactam ring during

its hydrolysis. As a result, its chemical identity changes, and its degradation product diketopiperazine-2', 5' (ADP), which is neither stable nor toxic, is produced (Gozlan et al., 2013, 2016; Lamm et al., 2009). This product continues to degrade into a few unstable and nontoxic degradation products, and one stable and toxic product, ADP3 (Gozlan et al., 2016; Lamm et al., 2009) (Fig. 3).

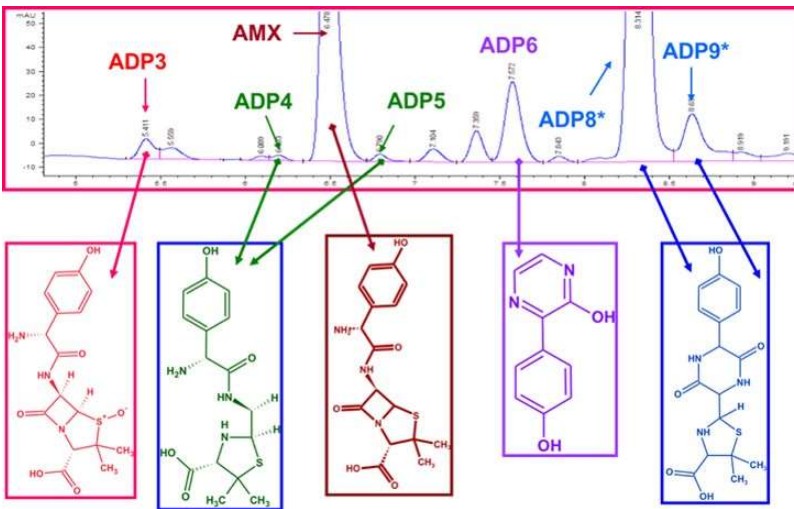

**Figure 3: AMX degradation products. ADP3 is the only stable and toxic degradation product (Avisar, 2017).**

ADP3 is the major degradation product of AMX and is defined as a stable, biologically active compound. It has been detected in surface water as well as in groundwater and represents its parent molecule's distribution. The case of AMX exemplifies the essence of degradation processes that might affect the parent compound and its degradation products while in the environment. It emphasizes the difficulty in detecting compounds and their degradation products in their stable form, which might lead to

the development of resistant bacteria and even cause other possible

health hazards to human, wild and domestic animals (Gozlan et al., 2010).



As already noted, there are many types of degradation products that might be more chemically stable than their parent compound (Gozlan et al., 2013). This stability is alarming as such products might form a chemically active structure that is toxic and threatens human and animal health. Another alarming outcome of these degradation products' chemical stability might be the development of resistant bacteria when the parent product is an antibiotic (Gozlan et al., 2010; Lamm et al.,

2009). Active degradation products might form during treatment by advanced oxidation processes (AOPs), although this is the suggested treatment type for the degradation of drug residues. Therefore, learning and investigating degradation processes under a variety of conditions, in addition to developing monitoring and measurement methods to detect them, is of great importance (Gozlan et al., 2010).

Effluent that has been allocated for agricultural irrigation might be rich in chemically stable drug residues and their degradation

products. These compounds then flow and drain into the environment, infiltrating with the water and mixing with surface and groundwater, especially next to cultivated areas (Avisar et al., 2009; Zhang et al., 2008). Measuring and detecting all degradation products in the environment is difficult since many have not yet been identified (by name or by chemical definition). Our knowledge is restricted and there are literally hundreds of unknown compounds in the aquatic environment that cannot be detected or measured (Lamm et al., 2009; Yin et al., 2017). We therefore cannot know whether these unknown

compounds are toxic or how their presence in the environment influences human health and the environment (Grossman and Rurman, 2007). Awareness of these dissolved and suspended organic compounds and their degradation products in water, and their stable structure which is not mineralized to water and CO2 during dedicated treatment (secondary, tertiary or AOPs), emphasize the importance of exploring the toxicity of these products in water.

### 3.3 How do residual drugs and their degradation products impact the irrigated environment?

By the time the drug residues reach the agroecosystem, via either irrigation or fertilizers using effluent or wastewater sludge, respectively, they may have gone through different processes that will seal their fate. Degradation (Grossberger et al., 2014), sorption and desorption to and from solid matter, infiltration into the groundwater (Paz et al., 2016) and plant uptake (Ben-Mordechay et al., 2017; Goldstein et al., 2014; Malchi et al., 2014) are examples of some of those processes. The processes are likely to rely upon the physicochemical properties of the pharmaceutical compounds—their molecular charge which

depends on the pKa, the water partition coefficient (Kow), and the soil properties—pH and total organic compounds (Goldstein et al., 2014).

Irrigation with effluent known to affect the soil properties, including physical, chemical and biological qualities of the soil (Dvorkin et al., 2012; Fernández-Gálvez et al., 2012; Ye et al., 2015). Especially concern is with their organic carbon (OC) content, that has been demonstrated to influence the processes or adsorption and desorption of the soils, and changes in the

soil's microbial community and its attendant activities (Rodríguez-Liébana et al., 2018).

In plant uptake, the drug residues and their degradation products are translocated to the leaves and fruit where they accumulate. They subsequently enter the food chain, as the agricultural products (leaves, fruit, and roots) are sold on the free market with no restrictions (Ben-Mordechay et al., 2017; Goldstein et al., 2014; Malchi et al., 2014).





In a different process, due to lipophilic sorption to the soil matrix resulting in high bioavailability, electrochemical interactions and an ion-trap effect, drug residues and their degradation products with changed chemical properties can be transported through the phloem. Therefore, not only do the degradation process and products need to be carefully elucidated, but also phloem transport and translocation need to be well deciphered (Ben-Mordechay et al., 2017; Christou et al., 2017; Trapp,

2000). A high concentration of drug residues (µg/kg), from hormones and antibiotics, among others, was found in wastewater sludge designated for agricultural field composting. These products are characterized by a high partition coefficient (Kow), meaning that they prefer the solid phase to the liquid phase. Therefore, when they leach and flow into the cultivated land, they absorb to the soil. Sometime later, when the conditions change, they desorb into the soil water, and are transported to the phloem or percolate into the groundwater (Shafrir and Avisar, 2012; Tenenbaum et al., 2014).

Drugs residues that have not been translocated to the vegetation may infiltrate from the agricultural fields into the saturated layer, and be pumped into production wells with the groundwater (Avisar et al., 2009; Lamm et al., 2009). Indirect, unknowing and uncontrolled consumption of drugs through drinking water and food can cause allergic reactions and health problems, thus constituting a serious human health threat (Christou et al., 2017).

**3.4 What are the regulated parameters in water regulation? Do these parameters define high-quality effluent?**

The worldwide approach for effluent use in agriculture is to combine risk assessment with the control of water-released diseases. This provides a framework for the development of health-based guidelines and standards. Three types of risk assessments are performed: microbial and chemical laboratory analyses, epidemiological studies, and quantitative microbial and chemical risk assessment (Carr et al., 2004). Much information is available from epidemiological studies of infectious disease transmission, as opposed to health risks from chemicals that are mostly based on quantitative risk assessment and

depend on the types of chemicals and the physical and chemical properties of the soil (Carr et al., 2004).

A variety of health-protective measures, including all customary chemical and microbial analyses, are carried out on a regular basis in all big treatment plants in Israel as well in most developed countries. In addition, measurements that attest to treatment quality in the treatment plants, such as BOD, total suspended solids (TSS), etc., are also tested on the products (Bower, 2000; Carr et al., 2004; Grossman and Rurman, 2007). In 2007, standards for effluent regulation were examined and unified. This

was followed by comprehensive studies performed in Israel by the standards committee ("Inbar" committee). The standard for effluent quality was unified with environmental and health implications, so that the effluent would be suitable for irrigation as well as for stream flow. That standard was integrated with the standard of quality for wastewater sludge produced in the treatment plant. Although many additional parameters were added in a discussion on the importance of their monitoring, the committee decided not to include micropollutants such as drug residues for either monitoring or tracing (Grossman and

Rurman, 2007). This is compatible with the global situation (Gerrity and Snyder, 2011). Except for monitoring and treatment efforts at the national level in Switzerland to remove antibiotics from the treatment-plant effluent, there are no regulatory restrictions regarding drug residues or their degradation products elsewhere (Swiss Federal Concil, 2016). In fact, legislators




in Israel still consider drug residues to be transparent. In other words, effluent containing resistant drug residues and many other degradation products is still considered high-quality effluent and allowed for any use in the environment.

Thus, drug residues and their degradation products in effluent are neither monitored nor measured, and there is no regulation restricting them. Nevertheless, there are many studies which testify to drug residues and their degradation products 5 contaminating soils, agricultural products and water resources, and being unknowingly consumed on a regular basis. The health implications and hazards are still under study, but the possible implications need to be consider. Drug residues mixing can synergistically enhance their toxicity, similarly to the phenomena happened with patients who taken mix of medications, although drug residue are on order of magnitude lower than administrated drugs. This scenario indicates a potential hazard and emphasizes the importance of legislation that will include these organic contaminants. Therefore, the proclaimed high quality 10 of the effluent produced in known treatment plants is not reliable.

### 4. Where do we go from here?

It is obvious that in the coming years, additional regulatory requirements will be needed to monitor and treat OMPs such as drug residues. The required treatment will be applied at the treatment plants, or at the wastewater source in the case of industries or hospitals (Gerrity and Snyder, 2011). Treatment at the wastewater source will need to be performed on smaller volumes and 15 with specific contaminants to degrade specific OMPs, to make the process more efficient and less expensive.

The main difficulty in degrading OMPs is their complex molecular structure. These compounds are only partially degraded by the common biological degradation process in treatment plants. The effluent leaving the plant is designated for agricultural irrigation with no restrictions, but it actually contains a mixture of OMPs. This hazard is already clear to researchers in the field of wastewater treatment, promoting many studies to prevent or eliminate this danger. In the last few years, research 20 groups worldwide have been developing combined methods to degrade and remove drug residues at the treatment plant as well as at the wastewater sources (industry and hospitals). The solutions stem from examinations of different technological aspects, the main focus being on the development and improvement of AOPs, which have been found to effectively treat the problem. The purpose of AOPs is to convert compounds into harmless or non-objectionable forms via a destructive process. During AOPs, the double bonds of the resistant compounds are broken by reaction with hydroxyl radicals (HO·) (Crittenden et al., 25 2012; Glaze et al., 1987). By the end of the AOP, in accordance with oxidizer and contaminant concentrations, the treated effluent contains the degradation products of the parent compounds, which are characterized by simpler chemical structures. The biological degradability of those degradation products is higher, enabling fast and efficient biological degradation. It is important to emphasize that not only are AOPs considered environmentally friendly (Oturan and Aaron, 2014), they have also been proven as such, and application of AOP technology has recently shown promising results in treatment plants (Avisar et 30 al., 2010; Lamm et al., 2009; Lester et al., 2012, 2013a; Zucker et al., 2015), the drug industry (Lester et al., 2013b) and hospitals (Avisar, 2017) in a few pioneering pilot platforms in Israel. The effluent produced in those platforms contained only negligible concentrations of OMPs.





### 5. Awareness

The most important task today is to raise awareness and perform cost–benefit analyses of the possible risks vs. the expense of new legislation and enforcement (Scott et al., 2004). Considering the many studies on this topic, cost–benefit analyses of OMP permits, monitoring, and enforcement are feasible, as is the potential implementation of a method to treat and eliminate them.

The authors think that "conducting business as usual", i.e., persisting with a regulatory attitude that ignores OMPs in effluent, is tantamount to "turning a blind eye" and allowing the overwhelming neglect to worsen. This is critical in the state of Israel, which is the leader in effluent reuse. Arguments that OMP concentrations are too small to influence human health are easy to make, but have never been proven. There have not yet been enough studies to confirm or reject this argument. It is therefore important to emphasize that even though drug residues and their degradation products are not supposed to be in the drinking

water at all, they do exist there, as a mixture. However, awareness to the possible danger impose by this unknown mixture must be carefully consider, before it can be ignored.

Globally, renewed legislation and the establishment of concentration permits for OMP contaminants in water resources and effluent are essential (Gozlan et al., 2014). Preparation for this essential stage includes comprehensive studies of the different chemical groups of the widespread drugs and their degradation products, method development to identify and measure them,

awareness of the analytical restrictions, and testing the possible influence of different OMP mixtures on human health. Similar source treatment of hospital and industrial wastewater is a must, and obliging the drug industry to treat their waste and establish specific permits for OMP concentrations in wastewater are critical to successfully implementing treatment methods to remove OMPs. This will enable the production of high-quality effluent that is readily available for any use. High-quality effluent following high-technology treatment is the target that Israel must strive for as a leading country in the use of effluent for

agricultural irrigation.

### 6. Conclusions

- It is important to be aware to the dangers depicted with the OMPs contains in regular effluent.
- The use of regular effluent for agriculture irrigation is one of the links caused to unconscious nondirective consumption of OMPs, either through agriculture products or by water resources contamination.

- Cost–benefit analyses of the possible risks vs. treatment expenses should be taken.
- A new legislation and enforcement regarding OMP contaminants in water resources and effluent are essential.

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

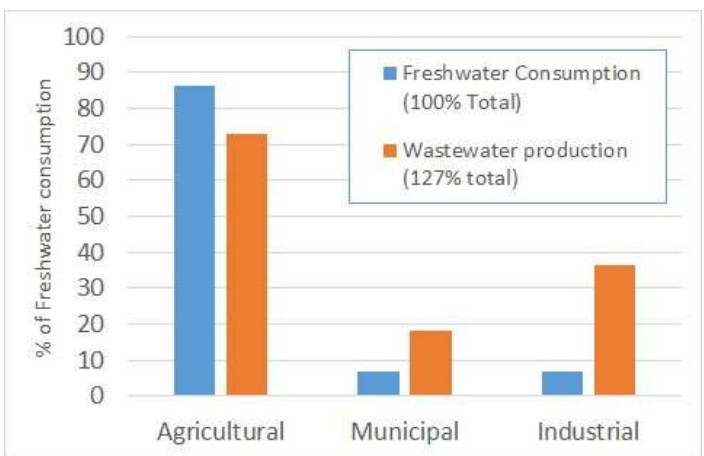

**Figure 1: Water consumption related to wastewater production (Data from UN-Water, 2017).**





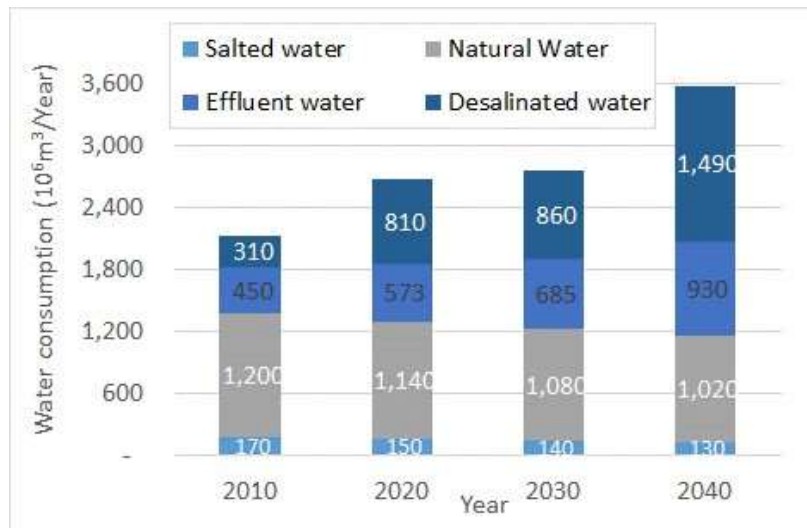

Figure 2: Estimates of water resource consumption up until 2040 in the Israeli water sector.

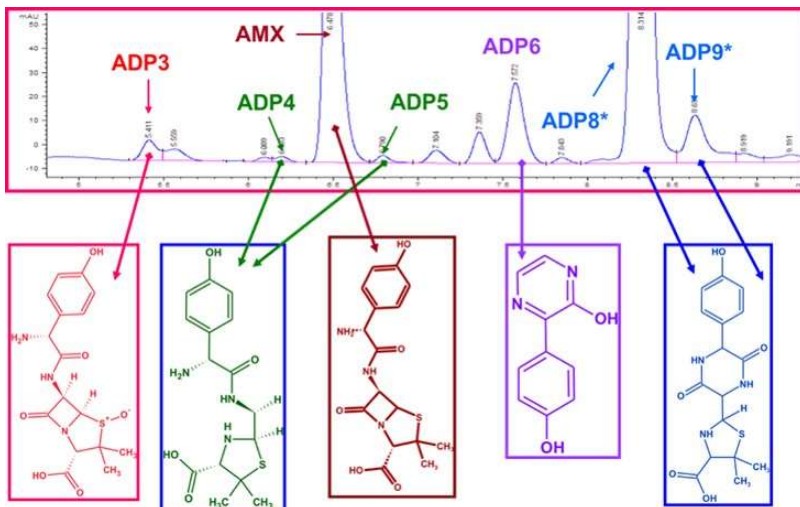

5    Figure 3: AMX degradation products. ADP3 is the only stable and toxic degradation product (Avisar, 2017).