# Peer review of "HESS Opinions: Agricultural irrigation with effluent – Pharmaceutical residues that we should worried about"

_Hydrology and Earth System Sciences, 2018_

## Referee Comment (RC1) · Anonymous Referee #1 · 3 Sep 2018

The manuscript by Avisar and Ronen-Eliraz raises few points for consideration regarding the need to monitor, regulate and treat drugs and their residuals from wastewater. The manuscript base the augments on the following statements: (i) The trends ofa wastewater generation and fresh water consumption is constantly growing leading to large amount of wastewater requiring treatment and shortage in fresh water resources in many parts of the world. (ii) Israel is the country with the highest rate of treated wastewater reuse and therefore can be considered a test case for treated wastewater reuse; and (iii) Drugs and their residues are persistent and are not degraded in conventual wastewater treatment facilities. After establishing this background the manuscript goes to several different directions including detection of drugs and their residues in

water resource and treatment of concentrated effluents. However there are few fundamental problems with this manuscript.

(1) The topic of this manuscript has been discussed at all levels for more than 2 decades now and the amount of work that has been published on each and every point that is reported in this manuscript is overwhelming. However, A reader without previous knowledge could understand that the drug residue in wastewater is a problem mainly in Israel and that most of the work has been done by two groups (20 out of 47 references or 38 scientific peer-reviewed publications) come from two group with the authors self-citing 14 of their his own publication).

(2) The manuscript does not bring any new perspective and many arguments are not fully supported. For example drug residues are considered contaminant of soil, water and sludge however it is also stated that their concentration are very low and that no proven health implication was reported to date. There are logical gaps and jumps between hot spots with high contamination that can be dealt by AOPs and low-level board contamination of soil and water that are not suitable for this type of treatment. Moreover, sometime the manuscript refer to drug and drug residues in other cases to pharmaceuticals and in other parts to organic micro-pollutants, which is much larger group. Finally the manuscript state that many of the pharmaceutical metabolites/transformation products are unknown and on the other hand calls for consideration of regulation these compounds.

(3) Relevant literature is missing for example: Mensingh J., Thurston C. 2015 PPCPs: Preparing For An Uncertain Regulatory Future. Water on line https://www.wateronline.com/doc/ppcps-preparing-for-an-uncertain-regulatory-future-0001 Boxall et al. 2012 Pharmaceuticals and Personal Care Products in the Environment: What Are the Big Questions? Environ Health Perspect. Sep; 120(9): 1221–1229. Ortiz de García, et al. 2013 Ranking of concern, based on environmental indexes, for pharmaceutical and personal care products: an application to the Spanish case. J Environ Manage.;129:384-97. doi:

10.1016/j.jenvman.2013.06.035. Epub 2013 Aug 28. Lamastra L, Balderacchi M, Trevisan M. 2016 Inclusion of emerging organic contaminants in groundwater monitoring plans.MethodsX. 2016 May 25;3:459-76. doi: 10.1016/j.mex.2016.05.008. eCollection 2016. EPA https://www.epa.gov/wqc/contaminants-emerging-concern-including-pharmaceuticals-and-personal-care-products G. Eckstein, 2012Comment: Emerging EPA Regulation of Pharmaceuticals in the Environment, 42 Envtl. L. Rep. 11105. (4) When one consider regulation a clear target and method must be suggested too. In this work no priority substance are suggested and it is noted twice in the text that to date the no toxicity at environmental levels was found. Moreover the detection methodology is very complex at best and often if one want to consider many metabolites the analytical methods are a subject for research and not routine analysis.

To conclude this manuscript is very limited in presenting the current literature (other than the authors own work) and instead provide a long list of clichés that are summarized by the "need for more research". No new perspective on the subject is provided. This is not enough to recommend publication.

Some specific comments:

Page 1 lines 26-27 - This is related mostly to biological contamination and waterborne diseases which are not really the subject of this work. This makes this statement a bit misleading. Page 3 fig 2 The amount of water resource will grow between 2010 and 2040 by ∼1400 Mcm/year and the reuse by only ∼500? Also is this original data/figure or should be credited to Bar-Eli, 2017 or someone else?

Page 5 lines 19-20 what were the measured ADP3 concentrations in that experiment?

Page 8 lines 4-5 This is very important point and should be discussed in details maybe even as a separated section. Please provide proofs for toxic effects and contamination of soil and water resources.

Page 8 lines 8-10 This is a problematic statement - the drug residues identity is often

unknown based on section 2 and concentrations of known active compounds are very low at ug/L or below with no proof that they cause health problem. Does these facts justify legislation that require very expensive and labor intensive analysis?

Page 8 section 4 In this section there is a mixture of point sources or hot spots like hospitals or manufacturing facilities with concentration effluents which are very different from contamination of groundwater, soils or municipal/domestic wastewater or even agricultural wastewater that were mentioned earlier.

Page 9 lines 7-8 You need to prove that the OMP residuals harmful and which specific elements are the problematic not the other way around.

Throughout the text: authors use OMP, drugs and pharmaceuticals referring to the same group of materials – please unify the terms. Check sub and super scripts.

---

## Referee Comment (RC2) · Anonymous Referee #2 · 3 Sep 2018

There are so many papers on this same topic that it is very hard to state something that goes beyond general statements (and we all do that). I learnt basically nothing. It is a nice (very nice) summary of many things that we already know, so, in my opinion, there is no novelty included. It reads like a proposal, introducing the problem (which is of course very relevant) but not yet producing any new scientific advance, rather than a promise to make it (new elements, metabolites, risk,...). For example, the introduction. What is the state of the art? Why do you write a paragraph such as the one lines 3-9, page 2? There are two paragraphs with just introduction to the problem, not to the literature. Then, pharmaceuticals appear in page 3, with the sentence: "Investigations of the occurrence, influence and toxicity of common organic contaminants, such

as fertilizers and oil compounds, among others, have also been published.". Exactly, this is what the paper is about, and I would like to see the relevant literature on the topic. And it goes well beyond pesticides. There are papers on the fate of drugs, UV filters, livestock antibiotics, illicit drugs, caffeine,... in surface water, groundwater and wastewater bodies. And there are hundreds of those. So, if I were a reader, I would expect a good state-of-the-art report that can help me understand the magnitude of the problem.

All three figures are irrelevant. The first is just plotting UN data; the second is local (Israel) case. None of them talk about pharmaceuticals (the title of the manuscript). The third one seems like writing again something from the literature, published by Grozlan et al (twice).

Now, for the specific questions you pose. I would definitely be very interested in a paper that addresses one (just one) of these questions thoroughly. It would probably look like an encyclopedia, though. But see how you deal with them: - What other components are present in effluents? The text talks only about carbamazepine and diclofenac . There are hundreds of organic molecules, with very varying concentration, and these concentration values depend on the degree of treatment (partially). Yes, carbamazepine is the most recalcitrant one, this is true.

- At what concentrations? There is only one sentence, and it could not be more general: "only a few nanograms to micrograms per liter". All of them? Everywhere? Is it relevant then?

- What are their degradation products? I believe that this question is not answered by looking at just one specific substance, that is already available in the literature. Plus, you do not explain what are the variables that condition the degradation paths that would really occur in a given case.

- How chemically stable, and how toxic are these degradation products? You do not address this problem, but only in one line saying a very general statement.

- How do these components and degradation products impact the irrigated environ-ment? So, how much gets into the plants eventually, how much remains sorbed in the soil, and how much goes to groundwater (where it mixes with billions of liters and fur-ther degrade due to the presence of soil biofilms that retain them and by the changes in redox conditions that degrade them).

- What are the regulated parameters in water regulation? Do they provide any pa-rameters in the context of recycled water? Do the regulated parameters indeed define high-quality effluents? This part I know nothing, and found the text very interesting, but it will not be enough to make it the core of the paper.

Altogether, I am sorry to say that I do not recommend this paper for publication.

---

## Referee Comment (RC3) · Anonymous Referee #3 · 19 Sep 2018

The manuscript introduces a topic that is largely discussed, but presents general statements that are poorly discussed and it's not clear the aim of this discussion. For a reader that is not familiar with the topic, it might give a general idea, but it does not go beyond that. Moreover, it does not present a comprehensive overview, but rather gathering different pieces on the topic. It appears like a regional assessment focused on Israel, but the title of the manuscript suggests something different. In fact, the topic the manuscript deals with, is presented by the authors as an issue that has been addressed mainly by Israeli researchers and is overweight of Israeli studies, of which a number are from the same authors of this work. Thus, it is very limited in presenting relevant and updated references. There are several works that could be cited, being

an issue that has caught a lot of attention worldwide in the last decades. In my opinion, the focus of the paper has to be better clarified and consequently the title and the abstract (and the manuscript too) modified to match the real focus of the paper.

The introduction and section 2 are dedicated specifically to water reuse and do not addresses the topic until section 3. Moreover, figure 1 and 2 (which, by the way lacks of a reference) do not contribute to the discussion, presenting irrelevant data regarding the topic of the manuscript. Half of section 3.1 is dedicated to describe general aspects of wastewater treatment rather than discussing the section's title, and the question remains unanswered ... which compounds at which concentrations? Carbamazepine and diclofenac ... which concentrations? Section 3.2, 3.3 and 3.4 are discussed poorly, for instance, half of section 3.4 discusses the conventional parameters present in water regulations (in Israel) just to say that pharmaceuticals are not regulated and regulation drug residues are required. Discussing in detail all the questions proposed could be too ambitious, but it would be more interesting discussing properly only one aspect, without losing focus. It's important to stress that reuse and emerging compounds are over debated topics in scientific community and therefore there are many papers, both reviews and research papers, dealing with these subjects.

Section 4: What about treatments (other than AOPs) for pharmaceutical residues removal and their efficiencies? It's not clear how discussing about AOPs application is strictly related to the topic defined in the title, also remembering that this processes are not the only option to remove emerging compounds. It's strange to dedicate a specific discussion on AOPs for emerging compounds removal, not discussing in deep the effect of biological process which is always present in a conventional wastewater treatment plant, meaning that current reuse practices are mainly referred to biologically-treated effluents.

Conclusions do not conclude anything and do not bring a new perspective, falling in commonplaces.

Moreover, sections 5 and 6 refer to OMPs rather than pharmaceuticals (what the manuscript is supposed to deal with according to the title). OMPs include compounds from sources other than pharmaceuticals.

Based on this remarks, in my opinion the manuscript is not recommended for publication.

---

## Author Comment (AC1) · 8 Nov 2018

Referee#1: (Q1-1) The topic of this manuscript has been discussed at all levels for more than 2 decades now and the amount of work that has been published on each and every point that is reported in this manuscript is overwhelming. However, A reader without previous knowledge could understand that the drug residue in wastewater is a problem mainly in Israel and that most of the work has been done by two groups (20 out of 47 references or 38 scientific peer-reviewed publications come from two group with the authors self-citing 14 of their his own publication).

(A1-1) Thank you for your comment. We definitely agree with the referee that some

implications of this topic has been discussed for more than 2 decades and therefore rewrote some of the paragraphs that might gave the wrong impression that this problem is only in Israel. Anyhow, we did emphasize the idea that on the last years, with the progress of the analytics abilities and the growing consumption of pharmaceutical, this subject become much more acute and though much popular topics with relevant implication to promote to the research frontier, as arise from the article. In addition, we add some more relevant references to give a broader representation of the many research group dealing with this subject.

(Q1-2) The manuscript does not bring any new perspective and many arguments are not fully supported. For example drug residues are considered contaminant of soil, water and sludge however it is also stated that their concentration are very low and that no proven health implication was reported to date. There are logical gaps and jumps between hot spots with high contamination that can be dealt by AOPs and low level board contamination of soil and water that are not suitable for this type of treatment. Moreover, sometime the manuscript refer to drug and drug residues in other cases to pharmaceuticals and in other parts to organic micro-pollutants, which is much larger group. Finally the manuscript state that many of the pharmaceutical metabolites/transformation products are unknown and on the other hand calls for consideration of regulation these compounds.

(A1-2) The article is an opinion article and its target was to emphasize the complications related to this topic that pronounced by those arguments mentioned by the referee. For example although the concentration of the drugs residues is low, they do consider contaminants. Health implication showed by some research but not on all of the researches that have been done; One of the complications is that there are hundreds of metabolites that are unknown but it doesn't means they are not toxic though regulation must be consider with this limitation according to the writer opinion. Besides, thank you for these remarks because it seems that the point was not clear so we rewrite it to be clearer. We add a reference that showed contamination of soil, water

and sludge, and reference that proven health complications to bridge on the gaps. In addition we broaden the section of treatment solution and add to the AOPs different kind of treatments dealing with the problem.

(Q1-3) Relevant literature is missing for example: Mensingh J., Thurston C. 2015 PPCPs: Preparing For An Uncertain Regulatory Future. Water on line https://www.wateronline.com/doc/ppcps-preparing-for-an-uncertain-regulatoryfuture-0001; Boxall et al. 2012 Pharmaceuticals and Personal Care Products in the Environment: What Are the Big Questions? Environ Health Perspect. Sep; 120(9): 1221–1229.; Ortiz de García, et al. 2013 Ranking of concern, based on environmental indexes, for pharmaceutical and personal care products: an application to the Spanish case. J Environ Manage. 129:384-97. doi: 10.1016/j.jenvman.2013.06.035. Epub 2013 Aug 28.; Lamastra L, Balderacchi M, Trevisan M. 2016 Inclusion of emerging organic ontaminants in groundwater monitoring plans. MethodsX. 2016 May 25;3:459-76. doi: 10.1016/j.mex.2016.05.008. eCollection 2016. EPA https://www.epa.gov/wqc/contaminants-emerging-concern-includingpharmaceuticals-and-personal-care-products; G. Eckstein, 2012Comment: Emerging EPA Regulation of Pharmaceuticals in the Environment, 42 Envtl. L. Rep. 11105.

(A1-3) Thank you very much for those relevant literature. We read it and broaden our references list.

(Q1-4) When one consider regulation a clear target and method must be suggested too. In this work no priority substance are suggested and it is noted twice in the text that to date the no toxicity at environmental levels was found. Moreover the detection methodology is very complex at best and often if one want to consider many metabolites the analytical methods are a subject for research and not routine analysis.

(A1-4) Thank you for your remark. We accept that the call for consideration of regulation is still not prepared and we should have written it differently: not as suggestion for regulation but in order to clear the point of this article - raise awareness. In order

to emphasize the complexity and the difficulties arise with this topic, connected with the complex methodology that goes with the water recycling (for detection and for removal), and the immediate need of recycling wastewater, it is important to be aware of the damages that might come with this as stated from the title.

Specific comments:

(Q1-5) Page 1 lines 26-27 - This is related mostly to biological contamination and waterborne diseases which are not really the subject of this work. This makes this statement a bit misleading. Page 3 fig 2 The amount of water resource will grow between 2010 and 2040 by _1400 Mcm/year and the reuse by only _500? Also is this original data/figure or should be credited to Bar-Eli, 2017 or someone else?

(A1-5) It is correct that this sentence is mostly related to biological contamination and waterborne diseases, and they are not the subject of this work, but it is part of the introduction which leads to the point that recycling water is not in a question but a present need and a fact. Therefore, solution, such as eliminate wastewater recycling, is unthinkable, although it fold in an unknown threat. It is correct that the water resources will grow by 1400Mcm/year and the effluent only by 500Mcm/year. The main water resource which grown so much is desalinated water. Anyway, the figure removed since we rewrite the paragraph and it was not relevant anymore.

(Q1-6) Page 5 lines 19-20 what were the measured ADP3 concentrations in that experiment?

(A1-6) ADP3 was given as example of degradation product of AMX. The sentence on line 19-20 is refer to AMX again as example of known pharmaceutical that degrade spontaneously in the aqueous environment to different metabolites (one of them is ADP3), that not all of them are known but some of them stable and toxic. In the context of this example it didn't seems interesting to discuss the actual concentration.

(Q1-7) Page 8 lines 4-5 This is very important point and should be discussed in de-

tails maybe even as a separated section. Please provide proofs for toxic effects and contamination of soil and water resources.

(A1-7) Thank you for this remark. It is important and we broaden the discussion on this important point in the text and added relevant references (Page 6, lines 5-20).

(Q1-8) Page 8 lines 8-10 This is a problematic statement - the drug residues identity is often known based on section 2 and concentrations of known active compounds are very low at ug/L or below with no proof that they cause health problem. Does these facts justify legislation that require very expensive and labor intensive analysis?

(A1-8) The drug residues identity in the wastewater or effluent is not always known. If it wrongly misunderstood on section 2 it is rewritten on section 2 clearer now. It stated in the article that although the concentration is very low, it might be dangerous. Thank to your referee, we understand that it might misunderstand though we rewrote the whole paragraph, and add relevant references. These facts are scary although the concentrations are very low, since accumulation of different discrete concentrations, might ended with significant concentrations, so as presented in the text, it is unthinkable to ignore the problem and leave it out of legislation.

(Q1-9) Page 8 section 4 In this section there is a mixture of point sources or hot spots like hospitals or manufacturing facilities with concentration effluents which are very different from contamination of groundwater, soils or municipal/domestic wastewater or even agricultural wastewater that were mentioned earlier.

(A1-9) Paragraph 4 deals with the possible solutions for pharmaceutical residues removal. Some of the solutions fold in the point sources treatment, which makes it easier to deal with. Those point source, like municipal wastewater or hospital waster, both contribute to the contamination of groundwater, though I am not sure what did the reviewer meant with this remark.

(Q1-10) Page 9 lines 7-8 You need to prove that the OMP residuals harmful and which

specific elements are the problematic not the other way around.

(A1-10) Thank you for the comment. We rewrite the sentence and add reference that showed the harmful impacts on the environment. In the specific lines mention above, it is part of the authors opinion to give a summary of how we think about neglecting the problem, even though, the specific target is sometimes unknown.

(Q1-11) Throughout the text: authors use OMP, drugs and pharmaceuticals referring to the same group of materials – please unify the terms. Check sub and super scripts.

(A1-11) Thank you for the comment. In this article we focus on pharmaceutical residues, which derivate from OMPs or/and from drugs, and so change it along the article to be uniform.

Please also note the supplement to this comment:
https://www.hydrol-earth-syst-sci-discuss.net/hess-2018-411/hess-2018-411-AC1-supplement.pdf

**Supplement:**

**HESS Opinions: Agricultural irrigation with effluent – Pharmaceutical residues that we should worried about**

Dror Avisar[1], Gefen Ronen-Eliraz[1]

[1] The Water Research Center, Hydrochemistry Research Group, Porter School of Environment and Earth Sciences, Faculty of Exact Sciences, Tel Aviv University, Tel Aviv 69978, Israel

*Correspondence to*: Dror Avisar (droravi@post.tau.ac.il)

**Abstract.** Policy regarding effluent water and reclamation aims to prevent environmental pollution while proposing an alternative water resource. Water makes up 99–99.9% of raw wastewater. Thus extracting organic and inorganic matter from water is a must. Worldwide, but especially in developed countries, great effort has been made to reuse wastewater, and it is becoming a reliable alternative source. Israel is the world leader in water reuse, allocating 85% of effluent water for agricultural irrigation. As such, it constitutes a "living laboratory" in which to study the implications of the intensive use of treated wastewater for agricultural irrigation, leading to research and legislation regarding effluent quality and regulation. Effluent produced in Israel is subject to severe regulations and standards and is considered suitable for every use except drinking water. It is mostly allocated for agricultural irrigation with no restrictions. The irrigated lands are close to natural water sources, and therefore water leaching from the fields infiltrate those sources, becoming part of the water cycle. A group of persistent and toxic nano- and micro-organic contaminants, including pharmaceutical residues, flows to water-treatment plants from hospitals, industry, agriculture and especially the domestic sector. These contaminants' chemical structure, characterized by a couple of aromatic rings and double bonds, makes them especially persistent; they are resistant to conventional biological treatment, used as a secondary treatment. As a result, the effluent that leaves the treatment plants, which is considered to be of high quality, actually contains pharmaceutical residues. After secondary and tertiary treatment, these persistent chemical residues can still be found in surface water, groundwater and agricultural products. Pharmaceutical residues in effluent allocated for agricultural irrigation are undesirable. Expansion of the monitoring system for those contaminants, improvement of the tertiary treatment, and implementation of advanced technologies for decomposition and removal of pharmaceutical contaminants are thus needed.

**1. Background**

Water scarcity is a global problem caused by increasing demand from one hand and dwindling resources, triggered both naturally and artificially (humanmade), on the other hand. The alert of "water scarcity" reflected in the World Health Organization (WHO) reports. According to WHO data, over 800,000 deaths worldwide were caused by contaminated drinking water, inadequate handwashing facilities, and inappropriate sanitation services, in 2012, and more than 30% of the world's population had no access to safely managed water (Ghebreyesus and Lake, 2017; Grojec, 2017; UN-Water, 2017). Therefore, alternative water resources are essential.

On the contrary, wastewater production keeps increasing, since most activities that use water produce wastewater in even higher quantities. Wastewater composed of roughly 99% water and 1% suspended, colloidal and dissolved solids. Untreated wastewater or wastewater without adequate treatment is a source of chemical, physical and biological pollution, threating on the environment and human health. (UN-Water, 2017). Therefore, opportunities offered by improved wastewater management and reuse is of a good water alternative resolving water scarcity and eliminate the threat of contamination that might be caused by untreated wastewater.

**2. Global water reuse**

On comprehensive worldwide assessments, 20% of wastewater going under some treatment (UN-Water, 2017). 85% of secondary effluent is reuse, allocated mainly for agricultural irrigation. Europe countries reuse about 2.4% of the treated wastewater, Mediterranean countries in Europe reuse 5%-12% of the treated wastewater except for Spain, which reuse about 25% of the treated wastewater. (Saliba et al., 2018; Sato et al., 2013). In this context, Israel is a leading country with 85% reuse of the treated wastewater (Cohen et al., 2016).

**3. The complications of wastewater reuse**

The quality of the treated wastewater is variable. It is composed of many different compounds depending on its source (municipal, industrial or agricultural) and wastewater-treatment plant operation and technology (Salgot & Folch, 2018). In contrast to fresh water, wastewater contained organic and inorganic contaminants, e.g., pharmaceutical or toxic metals, as dissolved and suspended particles (Grossman and Rurman, 2007; Strauch, 2011; WHO, 2006; Yeh et al., 2015; Zhang et al., 2013). Recycling of treated wastewater can lead to secondary contamination of soil and vegetation, surface water and adjacent groundwater because of these additional chemical compounds (Becerra-Castro, 2015; Christou et al., 2017; Desbiolles et al., 2018; WHO, 2006). On the last decades, contamination from pharmaceutical residues is noticed all over the world and therefore considered as emerging environmental pollution (Bagheri et al., 2016; Ortiz de Garcia et al., 2013; Strauch, 2011). Although pharmaceutical residues found in many aqueous environments worldwide, Israel, as a leading leading country in allocating treated wastewater for agricultural irrigation, which is in essence as a "living laboratory", and committed to carefully examining the various issues and implications that are tightly bound to unlimited wastewater reuse (Cohen et al., 2016).

In this paper, we examine the complexity of the implications arising **__from recycling wastewater,__** which contains discrete amounts of pharmaceutical residues and their metabolites, that should treated cautiously. To clarify these implications, we will answer the following questions: What other components are present in effluents? At what concentrations? What are their degradation products? How chemically stable, and how toxic are these degradation products? How do these components and degradation products affect the irrigated environment? What are the regulated parameters in water regulation? Do they provide any parameters in the context of recycled water? Do the regulated parameters indeed define high-quality effluents?

**3.1 What components are present in high-grade effluents? At what concentrations?**

The wastewater flowing into wastewater plants carries pharmaceutical residues, includingantibiotics, anti-inflammatory drugs, anti-depressants, sex hormones, lipid regulators and beta-blockers (Desbiolles et al., 2018). Depending on the specific circumstances, studies have shown that about 30–90% of all medications consumed, excreted via the urine and feces, as the parent compound or its metabolites, are introduced to wastewater (Adamczak et al., 2012; Christou et al., 2017).

The conventional treatment procedure for wastewater includes pretreatment, which is filtration and sedimentation, a secondary treatment that is based on biological degradation, and a tertiary treatment that consists of filtration and disinfection. The secondary treatment is the main stage designed to remove dissolved organic compounds. Apparently, it may remove -20% - +80% of the degraded organic compounds, as measured by biological oxygen demand (BOD) (Grandclément et al., 2017). In practice, at the end of the treatment procedure, the treated wastewater is considered "high-quality effluent" as it meets the standards for the main tested parameters, including BOD, COD, TSS, Soluble sodium percentage, Turbidity, N Total, $NO_3^-$, $NH_4^+$, Oil and Grease, etc. (Shakir et al., 2016). Studies have shown that during the biological treatment of wastewater, resistant pharmaceutical, for example ATZ, diazinon (DZN), diclofenac (DCF), carbamazepine (CBZ), metoprolol (METOP), do not degrade easily (Grandclément et al., 2017) and practically, different pharmaceutical residues have been detected in the aquatic environment around the world (Comber et al., 2018; Yang et al., 2017). The concentration of pharmaceutical residues in wastewater range from 100 to 100000 ng/L or even microgram/L in treated wastewater (Desbiolles et al., 2018; Lamm et al., 2009; Shafrir and Avisar, 2012; Souza et al., 2018; Zhang et al., 2008).

**3.2 Degradation products – Are they chemically stable? Are they toxic?**

Despite their chemical stability, some of the pharmaceutical residues might be degraded while flowing with the wastewater (Kosjak & Heath, 2011; Wang and Wang, 2016). In addition, natural degradation can caused by solar radiation, wastewater acidity (pH), or integration of additional organic compounds such as humic acid (Gozlan et al., 2010, 2014). Variation in wastewater composition influences degradation processes differently, although the parent compound will degrade preferentially to different specific degradation products under different conditions. This further complicates the monitoring and analysis of degradation products (Gozlan et al., 2013, 2016). The following example with the degradation of Amoxicillin (AMX) is emphasize part of the complications. AMX is the active ingredient in commonly consumed antibiotics worldwide. Surprisingly, it has never been detected in wastewater or effluent. It was found that AMX degrades easily due to opening of its strained β-lactam ring during its hydrolysis. As a result, its chemical identity changes, and its degradation product diketopiperazine-2', 5' (ADP), which is neither stable nor toxic, is produced (Gozlan et al., 2013, 2016; Lamm et al., 2009). This product continues to degrade into a few unstable and nontoxic degradation products, and one stable and toxic product, ADP3 (Gozlan et al., 2016; Lamm et al., 2009).

ADP3 is the major degradation product of AMX and is defined as a stable, biologically active compound. It has been detected in surface water as well as in groundwater and represents its parent molecule's distribution. The case of AMX exemplifies the essence of degradation processes that might affect the parent compound and its degradation products while in the environment. It emphasizes the difficulty in detecting compounds and their degradation products in their stable form, which might lead to the development of resistant bacteria and even cause other possible health hazards to human, wild and domestic animals (Gozlan et al., 2010).

As already noted, there are many types of degradation products that might be more chemically stable than their parent compound (Gozlan et al., 2013). This stability is alarming since such products might form a chemically active structure that is toxic and threatens human and animal health. Another alarming outcome of these degradation products' chemical stability might be the development of resistant bacteria when the parent product is an antibiotic (Gozlan et al., 2010; Lamm et al., 2009). Active degradation products might form during treatment by advanced oxidation processes (AOPs), although this is the suggested treatment type for the degradation of drug residues. Therefore, learning and investigating degradation processes under a variety of conditions, in addition to developing monitoring and measurement methods to detect them, is of great importance (Gozlan et al., 2010).

Effluent that has been allocated for agricultural irrigation might be rich in chemically stable drug residues and their degradation products. These compounds then flow and drain into the environment, infiltrating with the water and mixing with surface and groundwater, especially next to cultivated areas (Avisar et al., 2009; Zhang et al., 2008). Measuring and detecting all degradation products in the environment is difficult since many have not yet been identified (by name or by chemical definition). Our knowledge is restricted and there are literally hundreds of unknown compounds in the aquatic environment that cannot be detected or measured (Lamm et al., 2009; Yin et al., 2017), and therefore cannot know whether these unknown compounds are toxic or how their presence in the environment influences human health and the environment (Grossman and Rurman, 2007). On the other hand, we do know that pharmaceutical residues in the influent flowing into wastewater treatment plants may exert toxic or inhibitory effects on activated sludge bacteria (Yang et al., 2017). There are also studies that showed that long-term exposure to very-low concentration of pharmaceutical residues in the aquatic environment pose risk on the aquatic ecosystem (Strauch, 2011), and mixture of pharmaceutical residues and accumulation in soils, can threaten the environmental and human health (Beccera-Castro et al., 2015). Awareness to these dissolved and suspended organic compounds and their degradation products in water, and their stable structure which is not mineralized to water and $CO_2$ during dedicated treatment (secondary, tertiary or AOPs), emphasize the importance of exploring the toxicity of these products in water.

**3.3 How do residual drugs and their degradation products affect the irrigated environment?**

[revised manuscript text omitted]

5 In most of the world, pharmaceutical residues and their degradation products found in treated wastewater and has no regulation restricting them (Blair et al., 2015; Grandclément et al., 2017; Yang et al., 2017). It is particularly because of their low concentration, their large variety of chemical structures and the unknown degradation products, that posing a challenge on the analytical methods and though leave them out of regulation (Beccera-Castro et al., 2015; Blair et al., 2015). On the last decades, with analytical development and increasingly frequent medical prescriptions, and medication consumption, many studies

10 detected pharmaceutical residues and their metabolite, in contaminating soils (Blair et al., 2015), agricultural products (Becerra-Castro et al., 2015; Christou et al., 2017) and water resources (Ortiz de Garcia et al., 2013; Strauch 2011). It is already clear that some pharmaceutical residues cause health implications and hazards and the research must deepen the understanding of this context. Pharmaceutical residues mixing can synergistically enhance their toxicity, similarly to the phenomena happened with patients who taken mix of medications, although the residues are on order of magnitude lower than

15 administrated pharmaceutical (Comber et al., 2018; Ebele et al., 2017). It is also clear that the possible implications need to be consider for future regulation although there is much more we do not know than what we know (Desbiolles et al., 2018). The consideration of pharmaceutical residues in wastewater effluent got expression recently, while some pharmaceutical residues start to be monitored by European counties in the field of water policies. In the national level of Switzerland and England, monitoring and treatment efforts to remove or dilute antibiotics from the treatment-plant effluent, has already held (Comber et

20 al., 2018; Swiss Federal Concil, 2016).

**4. Where do we go from here?**

It is obvious that in the coming years, additional regulatory requirements will be needed to monitor and treat pharmaceutical residues. The required treatment will be applied at the treatment plants, or at the wastewater source in the case of industries or

25 hospitals (Gerrity and Snyder, 2011). Treatment at the wastewater source will need to be performed on smaller volumes and with specific contaminants to degrade specific pharmaceutical, to make the process more efficient and less expensive.

The main difficulty in degrading pharmaceutical is their physico-chemical properties. As mention earlier, these compounds are only partially degraded by the common biological degradation process  and the effluent leaving the plant designated for agricultural irrigation with no restrictions, is actually contains a mixture of pharmaceutical residues. Many studies promoting

30 researches to prevent or eliminate pharmaceutical residues to get to the effluent water. In the last few years, research groups worldwide have been developing combined methods to degrade and remove pharmaceutical residues at the treatment plant as well as at the wastewater sources (industry and hospitals. Biological treatments such as activated sludge, membrane bioreactor

treatment (MBR) are examples of biological degradation in treatment plants, in which operating conditions effect removal efficiency (Ejhed et al., 2018; Grandclément et al., 2017). Membrane filtration such as reverse osmosis reactor is also one of the solution under research (Cerro-Lopez et al., 2019). On the nanomaterials techniques there are metal nanoparticles, carbon nanotubes and nano filters which all of them found to be efficient removal for some of the contaminants (Bagheri et al., 2016).

5  One of the solutions stem from examinations of different technological aspects, focus on the development and improvement of Advanced Oxidation Process (AOPs). Some of AOPs techniques, can effectively treat the problem and degrade pharmaceutical residues, although not all of them (Lester et al., 2013; Siedlecka et al., 2018), Anyway, it comes up from different studies, that almost all of the treatment possibilities describe above, are often not sufficient ensure high removal of the pharmaceutical and therefore much of the technique improves by combining two or more treatment processes to one hybrid

10  treatment procedure (Grandclemente et al., 2017; Zucker et al., 2015).

**5. Awareness**

The most important task from the above manuscript is to raise awareness of the possible risks folds in using secondary effluent for agriculture irrigation. Arguments that pharmaceutical residues concentrations are too small to influence human health are

15  easy to make, but have never been proven. There have not yet been enough studies to confirm or reject this argument. It is therefore important to emphasize that even though pharmaceutical residues and their degradation products, are not supposed to be in the drinking water at all, they do exist there, as a mixture. However, awareness to the possible danger impose by this unknown mixture must be carefully consider, before it can be ignored.. The authors think that "conducting business as usual", i.e., persisting with a regulatory attitude that ignores OMPs in effluent, is tantamount to "turning a blind eye" and allowing the

20  overwhelming neglect to worsen.

Globally, renewed legislation and the establishment of concentration permits for pharmaceutical residues contaminants in water resources and effluent are essential (Gozlan et al., 2014). Preparation for this essential stage includes comprehensive studies of the different chemical groups of the widespread drugs and their degradation products, method development to identify and measure them, awareness of the analytical restrictions, and testing the possible influence of different

25  pharmaceutical residues mixtures on human health. Similar source treatment of hospital and industrial wastewater is a must, and obliging the drug industry to treat their waste and establish specific permits for pharmaceutical residues concentrations in wastewater are critical to successfully implementing treatment methods to remove it. This will enable the production of high-quality effluent that is readily available for any use.

---

## Author Comment (AC2) · 8 Nov 2018

Referee#2:

(Q2-1) There are so many papers on this same topic that it is very hard to state something that goes beyond general statements (and we all do that). I learnt basically nothing. It is a nice (very nice) summary of many things that we already know, so, in my opinion, there is no novelty included. It reads like a proposal, introducing the problem (which is of course very relevant) but not yet producing any new scientific advance, rather than a promise to make it (new elements, metabolites, risk). For example, the introduction. What is the state of the art? Why do you write a paragraph such as the

one lines 3-9, page 2? There are two paragraphs with just introduction to the problem, not to the literature. Then, pharmaceuticals appear in page 3, with the sentence: "Investigations of the occurrence, influence and toxicity of common organic contaminants, such as fertilizers and oil compounds, among others, have also been published. Exactly, this is what the paper is about, and I would like to see the relevant literature on the topic. And it goes well beyond pesticides. There are papers on the fate of drugs, UV filters, livestock antibiotics, illicit drugs, caffeine, in surface water, groundwater and wastewater bodies. And there are hundreds of those. So, if I were a reader, I would expect a good state-of-the-art report that can help me understand the magnitude of the problem.

(A2-1) The review on the manuscript arise some points that probably missed and therefore thank you for this referee, while according to the above remark we focus our manuscript as follow. We rewrite the introduction to lead to the main topic of this manuscript including background of the problem, and the state of the art of this topic (Page 1, Lines 25-31; Page 2, lines 1-12). We also added relevant literature and references. Along the text, we focus on the pharmaceutical residues, their sources, fate and where are they found in the environment.

(Q2-2) All three figures are irrelevant. The first is just plotting UN data; the second is local (Israel) case. None of them talk about pharmaceuticals (the title of the manuscript). The third one seems like writing again something from the literature, published by Gozlan et al (twice).

(A2-2) We accept this comment and remove the figures.

(Q2-3) Now, for the specific questions you pose. I would definitely be very interested in a paper that addresses one (just one) of these questions thoroughly. It would probably look like an encyclopedia, though. But see how you deal with them: - What other components are present in effluents? The text talks only about carbamazepine and diclofenac. There are hundreds of organic molecules, with very varying concentration,

and these concentration values depend on the degree of treatment (partially). Yes, carbamazepine is the most recalcitrant one, this is true.

(A2-3) Thank you for the remark. We added some more examples for other pharmaceutical found in treated wastewater and their references (Page 3, Lines 4-6; page 3, lines 13-20).

(Q2-4) At what concentrations? There is only one sentence, and it could not be more general: "only a few Nano grams to micrograms per liter". All of them? Everywhere? Is it relevant then?

(A2-4) As it described in the manuscript, the concentrations found is varied and it depend on the specific physico-chemical properties of the specific pharmaceutical residue as well as on the wastewater-treatment plant operation and technology (Page 3, Line 14). Yes, it is very relevant because in this manuscript we asked to present the complication of this subject which part of it is due to the variety of the products and their concentration specifically in treated wastewater.

(Q2-5) What are their degradation products? I believe that this question is not answered by looking at just one specific substance, that is already available in the literature. Plus, you do not explain what are the variables that condition the degradation paths that would really occur in a given case.

(A2-5) The manuscript we introduced is an opinion and though on many of the point we present we gave an example but did not reviewed all of the possibilities. However, in order to make the point clearer we highlight the complexion arise from this example (Page 3, Lines 21-32).

(Q2-6) How chemically stable, and how toxic are these degradation products? You do not address this problem, but only in one line saying a very general statement.

(A2-6) Than you for your comment. We rewrote the paragraph and gave more example and references (Page 4, Lines 8-13; page 4, lines 20-27).

(Q2-7) How do these components and degradation products impact the irrigated environment? So, how much gets into the plants eventually, how much remains sorbed in the soil, and how much goes to groundwater (where it mixes with billions of liters and further degrade due to the presence of soil biofilms that retain them and by the changes in redox conditions that degrade them).

(A2-7) Thank you for this comment. It is a big issue and study by different groups around the world deeply discuss this subject. For our point of you, we present the possible impacts of the pharmaceutical residues and their degradation products on the irrigated environment generally and add references that discuss the implications of specific compound or environment (Page 5, lines 5-25). We did not discuss the mechanisms or the diversion of the impact on the environment since this is behind the scope of this manuscript.

(Q2-8) What are the regulated parameters in water regulation? Do they provide any parameters in the context of recycled water? Do the regulated parameters indeed define high-quality effluents? This part I know nothing, and found the text very interesting, but it will not be enough to make it the core of the paper.

(A2-8) The parameter that are regulated in water regulation including BOD, COD, TSS, Soluble sodium percentage, Turbidity, N Total, NO3-, NH4+, Oil and Grease, etc., are examine in the context of recycled water in the same manner (Page 6, lines 1-5). The meaning is that when effluent is checked for those parameters and the results are in the range of regulations, the effluent consider "high quality effluent".

Please also note the supplement to this comment:
https://www.hydrol-earth-syst-sci-discuss.net/hess-2018-411/hess-2018-411-AC2-supplement.pdf
* * *
[Figure]

**Supplement:**

**HESS Opinions: Agricultural irrigation with effluent – Pharmaceutical residues that we should worried about**

Dror Avisar[1], Gefen Ronen-Eliraz[1]

[1] The Water Research Center, Hydrochemistry Research Group, Porter School of Environment and Earth Sciences, Faculty of Exact Sciences, Tel Aviv University, Tel Aviv 69978, Israel

*Correspondence to*: Dror Avisar (droravi@post.tau.ac.il)

**Abstract.** Policy regarding effluent water and reclamation aims to prevent environmental pollution while proposing an alternative water resource. Water makes up 99–99.9% of raw wastewater. Thus extracting organic and inorganic matter from water is a must. Worldwide, but especially in developed countries, great effort has been made to reuse wastewater, and it is becoming a reliable alternative source. Israel is the world leader in water reuse, allocating 85% of effluent water for agricultural irrigation. As such, it constitutes a "living laboratory" in which to study the implications of the intensive use of treated wastewater for agricultural irrigation, leading to research and legislation regarding effluent quality and regulation. Effluent produced in Israel is subject to severe regulations and standards and is considered suitable for every use except drinking water. It is mostly allocated for agricultural irrigation with no restrictions. The irrigated lands are close to natural water sources, and therefore water leaching from the fields infiltrate those sources, becoming part of the water cycle. A group of persistent and toxic nano- and micro-organic contaminants, including pharmaceutical residues, flows to water-treatment plants from hospitals, industry, agriculture and especially the domestic sector. These contaminants' chemical structure, characterized by a couple of aromatic rings and double bonds, makes them especially persistent; they are resistant to conventional biological treatment, used as a secondary treatment. As a result, the effluent that leaves the treatment plants, which is considered to be of high quality, actually contains pharmaceutical residues. After secondary and tertiary treatment, these persistent chemical residues can still be found in surface water, groundwater and agricultural products. Pharmaceutical residues in effluent allocated for agricultural irrigation are undesirable. Expansion of the monitoring system for those contaminants, improvement of the tertiary treatment, and implementation of advanced technologies for decomposition and removal of pharmaceutical contaminants are thus needed.

**1. Background**

Water scarcity is a global problem caused by increasing demand from one hand and dwindling resources, triggered both naturally and artificially (humanmade), on the other hand. The alert of "water scarcity" reflected in the World Health Organization (WHO) reports. According to WHO data, over 800,000 deaths worldwide were caused by contaminated drinking water, inadequate handwashing facilities, and inappropriate sanitation services, in 2012, and more than 30% of the world's population had no access to safely managed water (Ghebreyesus and Lake, 2017; Grojec, 2017; UN-Water, 2017). Therefore, alternative water resources are essential.

On the contrary, wastewater production keeps increasing, since most activities that use water produce wastewater in even higher quantities. Wastewater composed of roughly 99% water and 1% suspended, colloidal and dissolved solids. Untreated wastewater or wastewater without adequate treatment is a source of chemical, physical and biological pollution, threating on the environment and human health. (UN-Water, 2017). Therefore, opportunities offered by improved wastewater management and reuse is of a good water alternative resolving water scarcity and eliminate the threat of contamination that might be caused by untreated wastewater.

**2. Global water reuse**

On comprehensive worldwide assessments, 20% of wastewater going under some treatment (UN-Water, 2017). 85% of secondary effluent is reuse, allocated mainly for agricultural irrigation. Europe countries reuse about 2.4% of the treated wastewater, Mediterranean countries in Europe reuse 5%-12% of the treated wastewater except for Spain, which reuse about 25% of the treated wastewater. (Saliba et al., 2018; Sato et al., 2013). In this context, Israel is a leading country with 85% reuse of the treated wastewater (Cohen et al., 2016).

**3. The complications of wastewater reuse**

The quality of the treated wastewater is variable. It is composed of many different compounds depending on its source (municipal, industrial or agricultural) and wastewater-treatment plant operation and technology (Salgot & Folch, 2018). In contrast to fresh water, wastewater contained organic and inorganic contaminants, e.g., pharmaceutical or toxic metals, as dissolved and suspended particles (Grossman and Rurman, 2007; Strauch, 2011; WHO, 2006; Yeh et al., 2015; Zhang et al., 2013). Recycling of treated wastewater can lead to secondary contamination of soil and vegetation, surface water and adjacent groundwater because of these additional chemical compounds (Becerra-Castro, 2015; Christou et al., 2017; Desbiolles et al., 2018; WHO, 2006). On the last decades, contamination from pharmaceutical residues is noticed all over the world and therefore considered as emerging environmental pollution (Bagheri et al., 2016; Ortiz de Garcia et al., 2013; Strauch, 2011). Although pharmaceutical residues found in many aqueous environments worldwide, Israel, as a leading leading country in allocating treated wastewater for agricultural irrigation, which is in essence as a "living laboratory", and committed to carefully examining the various issues and implications that are tightly bound to unlimited wastewater reuse (Cohen et al., 2016).

In this paper, we examine the complexity of the implications arising **__from recycling wastewater,__** which contains discrete amounts of pharmaceutical residues and their metabolites, that should treated cautiously. To clarify these implications, we will answer the following questions: What other components are present in effluents? At what concentrations? What are their degradation products? How chemically stable, and how toxic are these degradation products? How do these components and degradation products affect the irrigated environment? What are the regulated parameters in water regulation? Do they provide any parameters in the context of recycled water? Do the regulated parameters indeed define high-quality effluents?

**3.1 What components are present in high-grade effluents? At what concentrations?**

The wastewater flowing into wastewater plants carries pharmaceutical residues, includingantibiotics, anti-inflammatory drugs, anti-depressants, sex hormones, lipid regulators and beta-blockers (Desbiolles et al., 2018). Depending on the specific circumstances, studies have shown that about 30–90% of all medications consumed, excreted via the urine and feces, as the parent compound or its metabolites, are introduced to wastewater (Adamczak et al., 2012; Christou et al., 2017).

The conventional treatment procedure for wastewater includes pretreatment, which is filtration and sedimentation, a secondary treatment that is based on biological degradation, and a tertiary treatment that consists of filtration and disinfection. The secondary treatment is the main stage designed to remove dissolved organic compounds. Apparently, it may remove -20% - +80% of the degraded organic compounds, as measured by biological oxygen demand (BOD) (Grandclément et al., 2017). In practice, at the end of the treatment procedure, the treated wastewater is considered "high-quality effluent" as it meets the standards for the main tested parameters, including BOD, COD, TSS, Soluble sodium percentage, Turbidity, N Total, $NO_3^-$, $NH_4^+$, Oil and Grease, etc. (Shakir et al., 2016). Studies have shown that during the biological treatment of wastewater, resistant pharmaceutical, for example ATZ, diazinon (DZN), diclofenac (DCF), carbamazepine (CBZ), metoprolol (METOP), do not degrade easily (Grandclément et al., 2017) and practically, different pharmaceutical residues have been detected in the aquatic environment around the world (Comber et al., 2018; Yang et al., 2017). The concentration of pharmaceutical residues in wastewater range from 100 to 100000 ng/L or even microgram/L in treated wastewater (Desbiolles et al., 2018; Lamm et al., 2009; Shafrir and Avisar, 2012; Souza et al., 2018; Zhang et al., 2008).

**3.2 Degradation products – Are they chemically stable? Are they toxic?**

Despite their chemical stability, some of the pharmaceutical residues might be degraded while flowing with the wastewater (Kosjak & Heath, 2011; Wang and Wang, 2016). In addition, natural degradation can caused by solar radiation, wastewater acidity (pH), or integration of additional organic compounds such as humic acid (Gozlan et al., 2010, 2014). Variation in wastewater composition influences degradation processes differently, although the parent compound will degrade preferentially to different specific degradation products under different conditions. This further complicates the monitoring and analysis of degradation products (Gozlan et al., 2013, 2016). The following example with the degradation of Amoxicillin (AMX) is emphasize part of the complications. AMX is the active ingredient in commonly consumed antibiotics worldwide. Surprisingly, it has never been detected in wastewater or effluent. It was found that AMX degrades easily due to opening of its strained β-lactam ring during its hydrolysis. As a result, its chemical identity changes, and its degradation product diketopiperazine-2', 5' (ADP), which is neither stable nor toxic, is produced (Gozlan et al., 2013, 2016; Lamm et al., 2009). This product continues to degrade into a few unstable and nontoxic degradation products, and one stable and toxic product, ADP3 (Gozlan et al., 2016; Lamm et al., 2009).

ADP3 is the major degradation product of AMX and is defined as a stable, biologically active compound. It has been detected in surface water as well as in groundwater and represents its parent molecule's distribution. The case of AMX exemplifies the essence of degradation processes that might affect the parent compound and its degradation products while in the environment. It emphasizes the difficulty in detecting compounds and their degradation products in their stable form, which might lead to the development of resistant bacteria and even cause other possible health hazards to human, wild and domestic animals (Gozlan et al., 2010).

As already noted, there are many types of degradation products that might be more chemically stable than their parent compound (Gozlan et al., 2013). This stability is alarming since such products might form a chemically active structure that is toxic and threatens human and animal health. Another alarming outcome of these degradation products' chemical stability might be the development of resistant bacteria when the parent product is an antibiotic (Gozlan et al., 2010; Lamm et al., 2009). Active degradation products might form during treatment by advanced oxidation processes (AOPs), although this is the suggested treatment type for the degradation of drug residues. Therefore, learning and investigating degradation processes under a variety of conditions, in addition to developing monitoring and measurement methods to detect them, is of great importance (Gozlan et al., 2010).

Effluent that has been allocated for agricultural irrigation might be rich in chemically stable drug residues and their degradation products. These compounds then flow and drain into the environment, infiltrating with the water and mixing with surface and groundwater, especially next to cultivated areas (Avisar et al., 2009; Zhang et al., 2008). Measuring and detecting all degradation products in the environment is difficult since many have not yet been identified (by name or by chemical definition). Our knowledge is restricted and there are literally hundreds of unknown compounds in the aquatic environment that cannot be detected or measured (Lamm et al., 2009; Yin et al., 2017), and therefore cannot know whether these unknown compounds are toxic or how their presence in the environment influences human health and the environment (Grossman and Rurman, 2007). On the other hand, we do know that pharmaceutical residues in the influent flowing into wastewater treatment plants may exert toxic or inhibitory effects on activated sludge bacteria (Yang et al., 2017). There are also studies that showed that long-term exposure to very-low concentration of pharmaceutical residues in the aquatic environment pose risk on the aquatic ecosystem (Strauch, 2011), and mixture of pharmaceutical residues and accumulation in soils, can threaten the environmental and human health (Beccera-Castro et al., 2015). Awareness to these dissolved and suspended organic compounds and their degradation products in water, and their stable structure which is not mineralized to water and $CO_2$ during dedicated treatment (secondary, tertiary or AOPs), emphasize the importance of exploring the toxicity of these products in water.

**3.3 How do residual drugs and their degradation products affect the irrigated environment?**

[revised manuscript text omitted]

5 In most of the world, pharmaceutical residues and their degradation products found in treated wastewater and has no regulation restricting them (Blair et al., 2015; Grandclément et al., 2017; Yang et al., 2017). It is particularly because of their low concentration, their large variety of chemical structures and the unknown degradation products, that posing a challenge on the analytical methods and though leave them out of regulation (Beccera-Castro et al., 2015; Blair et al., 2015). On the last decades, with analytical development and increasingly frequent medical prescriptions, and medication consumption, many studies

10 detected pharmaceutical residues and their metabolite, in contaminating soils (Blair et al., 2015), agricultural products (Becerra-Castro et al., 2015; Christou et al., 2017) and water resources (Ortiz de Garcia et al., 2013; Strauch 2011). It is already clear that some pharmaceutical residues cause health implications and hazards and the research must deepen the understanding of this context. Pharmaceutical residues mixing can synergistically enhance their toxicity, similarly to the phenomena happened with patients who taken mix of medications, although the residues are on order of magnitude lower than

15 administrated pharmaceutical (Comber et al., 2018; Ebele et al., 2017). It is also clear that the possible implications need to be consider for future regulation although there is much more we do not know than what we know (Desbiolles et al., 2018). The consideration of pharmaceutical residues in wastewater effluent got expression recently, while some pharmaceutical residues start to be monitored by European counties in the field of water policies. In the national level of Switzerland and England, monitoring and treatment efforts to remove or dilute antibiotics from the treatment-plant effluent, has already held (Comber et

20 al., 2018; Swiss Federal Concil, 2016).

**4. Where do we go from here?**

It is obvious that in the coming years, additional regulatory requirements will be needed to monitor and treat pharmaceutical residues. The required treatment will be applied at the treatment plants, or at the wastewater source in the case of industries or

25 hospitals (Gerrity and Snyder, 2011). Treatment at the wastewater source will need to be performed on smaller volumes and with specific contaminants to degrade specific pharmaceutical, to make the process more efficient and less expensive.

The main difficulty in degrading pharmaceutical is their physico-chemical properties. As mention earlier, these compounds are only partially degraded by the common biological degradation process  and the effluent leaving the plant designated for agricultural irrigation with no restrictions, is actually contains a mixture of pharmaceutical residues. Many studies promoting

30 researches to prevent or eliminate pharmaceutical residues to get to the effluent water. In the last few years, research groups worldwide have been developing combined methods to degrade and remove pharmaceutical residues at the treatment plant as well as at the wastewater sources (industry and hospitals. Biological treatments such as activated sludge, membrane bioreactor

treatment (MBR) are examples of biological degradation in treatment plants, in which operating conditions effect removal efficiency (Ejhed et al., 2018; Grandclément et al., 2017). Membrane filtration such as reverse osmosis reactor is also one of the solution under research (Cerro-Lopez et al., 2019). On the nanomaterials techniques there are metal nanoparticles, carbon nanotubes and nano filters which all of them found to be efficient removal for some of the contaminants (Bagheri et al., 2016).

5  One of the solutions stem from examinations of different technological aspects, focus on the development and improvement of Advanced Oxidation Process (AOPs). Some of AOPs techniques, can effectively treat the problem and degrade pharmaceutical residues, although not all of them (Lester et al., 2013; Siedlecka et al., 2018), Anyway, it comes up from different studies, that almost all of the treatment possibilities describe above, are often not sufficient ensure high removal of the pharmaceutical and therefore much of the technique improves by combining two or more treatment processes to one hybrid

10  treatment procedure (Grandclemente et al., 2017; Zucker et al., 2015).

**5. Awareness**

The most important task from the above manuscript is to raise awareness of the possible risks folds in using secondary effluent for agriculture irrigation. Arguments that pharmaceutical residues concentrations are too small to influence human health are

15  easy to make, but have never been proven. There have not yet been enough studies to confirm or reject this argument. It is therefore important to emphasize that even though pharmaceutical residues and their degradation products, are not supposed to be in the drinking water at all, they do exist there, as a mixture. However, awareness to the possible danger impose by this unknown mixture must be carefully consider, before it can be ignored.. The authors think that "conducting business as usual", i.e., persisting with a regulatory attitude that ignores OMPs in effluent, is tantamount to "turning a blind eye" and allowing the

20  overwhelming neglect to worsen.

Globally, renewed legislation and the establishment of concentration permits for pharmaceutical residues contaminants in water resources and effluent are essential (Gozlan et al., 2014). Preparation for this essential stage includes comprehensive studies of the different chemical groups of the widespread drugs and their degradation products, method development to identify and measure them, awareness of the analytical restrictions, and testing the possible influence of different

25  pharmaceutical residues mixtures on human health. Similar source treatment of hospital and industrial wastewater is a must, and obliging the drug industry to treat their waste and establish specific permits for pharmaceutical residues concentrations in wastewater are critical to successfully implementing treatment methods to remove it. This will enable the production of high-quality effluent that is readily available for any use.

---

## Author Comment (AC3) · 8 Nov 2018

Referee#3:

(Q3-1) The manuscript introduces a topic that is largely discussed, but presents general statements that are poorly discussed and it's not clear the aim of this discussion. For a reader that is not familiar with the topic, it might give a general idea, but it does not go beyond that. Moreover, it does not present a comprehensive overview, but rather gathering different pieces on the topic. It appears like a regional assessment focused on Israel, but the title of the manuscript suggests something different. In fact, the topic the manuscript deals with, is presented by the authors as an issue that has been

addressed mainly by Israeli researchers and is overweight of Israeli studies, of which a number are from the same authors of this work. Thus, it is very limited in presenting relevant and updated references. There are several works that could be cited, being an issue that has caught a lot of attention worldwide in the last decades. In my opinion, the focus of the paper has to be better clarified and consequently the title and the abstract (and the manuscript too) modified to match the real focus of the paper.

(A3-1) Thank you for the comment. We rewrite the manuscript and added more references. It is now arrange to emphasize its point, which written in the manuscripts' title and present a comprehensive overview on the subject. We also wrote it now in away, to stress that the problem is over the whole world, and so many research group studied its implication.

(Q3-2) The introduction and section 2 are dedicated specifically to water reuse and do not addresses the topic until section 3. Moreover, figure 1 and 2 (which, by the way lacks of a reference) do not contribute to the discussion, presenting irrelevant data regarding the topic of the manuscript. Half of section 3.1 is dedicated to describe general aspects of wastewater treatment rather than discussing the section's title, and the question remains unanswered ... which compounds at which concentrations? Carbamazepine and diclofenac ... which concentrations? Section 3.2, 3.3 and 3.4 are discussed poorly, for instance, half of section 3.4 discusses the conventional parameters present in water regulations (in Israel) just to say that pharmaceuticals are not regulated and regulation drug residues are required. Discussing in detail all the questions proposed could be too ambitious, but it would be more interesting discussing properly only one aspect, without losing focus. It's important to stress that reuse and emerging compounds are over debated topics in scientific community and therefore there are many papers, both reviews and research papers, dealing with these subjects.

(A3-2) Thank you for the comments. We rewrote sections 1 and 2 to present the background and point we want to raise, right from the begging (Page 2, lines 25-31; Page 2, lines 1-12). We rewrote and arrange the sections 3.1 to 3.4 to focus in the

subject present in their title and add relevant reference (Page 3 line 3 – Page 6, line 20).

(Q3-3) Section 4: What about treatments (other than AOPs) for pharmaceutical residues removal and their efficiencies? It is not clear how discussing about AOPs application is strictly related to the topic defined in the title, also remembering that this processes are not the only option to remove emerging compounds. It is strange to dedicate a specific discussion on AOPs for emerging compounds removal, not discussing in deep the effect of biological process which is always present in a conventional wastewater treatment plant, meaning that current reuse practices are mainly referred to biologically treated effluents.

(A3-3) The author thank the reviewer for this remark. This section was rewritten and broaden, to present other treatments for the remove of pharmaceutical residues (Page 6, line 23 – Page 7, line 10).

Sincerely yours,

Prof. Dror Avisar Hydro-chemistry Director of the Water Research Center Porter School of the Environment and Earth Sciences Faculty of Exact Sciences Tel Aviv University

Please also note the supplement to this comment:
https://www.hydrol-earth-syst-sci-discuss.net/hess-2018-411/hess-2018-411-AC3-supplement.pdf
* * *
[Figure]

**Supplement:**

**HESS Opinions: Agricultural irrigation with effluent – Pharmaceutical residues that we should worried about**

Dror Avisar[1], Gefen Ronen-Eliraz[1]

[1] The Water Research Center, Hydrochemistry Research Group, Porter School of Environment and Earth Sciences, Faculty of Exact Sciences, Tel Aviv University, Tel Aviv 69978, Israel

*Correspondence to*: Dror Avisar (droravi@post.tau.ac.il)

**Abstract.** Policy regarding effluent water and reclamation aims to prevent environmental pollution while proposing an alternative water resource. Water makes up 99–99.9% of raw wastewater. Thus extracting organic and inorganic matter from water is a must. Worldwide, but especially in developed countries, great effort has been made to reuse wastewater, and it is becoming a reliable alternative source. Israel is the world leader in water reuse, allocating 85% of effluent water for agricultural irrigation. As such, it constitutes a "living laboratory" in which to study the implications of the intensive use of treated wastewater for agricultural irrigation, leading to research and legislation regarding effluent quality and regulation. Effluent produced in Israel is subject to severe regulations and standards and is considered suitable for every use except drinking water. It is mostly allocated for agricultural irrigation with no restrictions. The irrigated lands are close to natural water sources, and therefore water leaching from the fields infiltrate those sources, becoming part of the water cycle. A group of persistent and toxic nano- and micro-organic contaminants, including pharmaceutical residues, flows to water-treatment plants from hospitals, industry, agriculture and especially the domestic sector. These contaminants' chemical structure, characterized by a couple of aromatic rings and double bonds, makes them especially persistent; they are resistant to conventional biological treatment, used as a secondary treatment. As a result, the effluent that leaves the treatment plants, which is considered to be of high quality, actually contains pharmaceutical residues. After secondary and tertiary treatment, these persistent chemical residues can still be found in surface water, groundwater and agricultural products. Pharmaceutical residues in effluent allocated for agricultural irrigation are undesirable. Expansion of the monitoring system for those contaminants, improvement of the tertiary treatment, and implementation of advanced technologies for decomposition and removal of pharmaceutical contaminants are thus needed.

**1. Background**

Water scarcity is a global problem caused by increasing demand from one hand and dwindling resources, triggered both naturally and artificially (humanmade), on the other hand. The alert of "water scarcity" reflected in the World Health Organization (WHO) reports. According to WHO data, over 800,000 deaths worldwide were caused by contaminated drinking water, inadequate handwashing facilities, and inappropriate sanitation services, in 2012, and more than 30% of the world's population had no access to safely managed water (Ghebreyesus and Lake, 2017; Grojec, 2017; UN-Water, 2017). Therefore, alternative water resources are essential.

On the contrary, wastewater production keeps increasing, since most activities that use water produce wastewater in even higher quantities. Wastewater composed of roughly 99% water and 1% suspended, colloidal and dissolved solids. Untreated wastewater or wastewater without adequate treatment is a source of chemical, physical and biological pollution, threating on the environment and human health. (UN-Water, 2017). Therefore, opportunities offered by improved wastewater management and reuse is of a good water alternative resolving water scarcity and eliminate the threat of contamination that might be caused by untreated wastewater.

**2. Global water reuse**

On comprehensive worldwide assessments, 20% of wastewater going under some treatment (UN-Water, 2017). 85% of secondary effluent is reuse, allocated mainly for agricultural irrigation. Europe countries reuse about 2.4% of the treated wastewater, Mediterranean countries in Europe reuse 5%-12% of the treated wastewater except for Spain, which reuse about 25% of the treated wastewater. (Saliba et al., 2018; Sato et al., 2013). In this context, Israel is a leading country with 85% reuse of the treated wastewater (Cohen et al., 2016).

**3. The complications of wastewater reuse**

The quality of the treated wastewater is variable. It is composed of many different compounds depending on its source (municipal, industrial or agricultural) and wastewater-treatment plant operation and technology (Salgot & Folch, 2018). In contrast to fresh water, wastewater contained organic and inorganic contaminants, e.g., pharmaceutical or toxic metals, as dissolved and suspended particles (Grossman and Rurman, 2007; Strauch, 2011; WHO, 2006; Yeh et al., 2015; Zhang et al., 2013). Recycling of treated wastewater can lead to secondary contamination of soil and vegetation, surface water and adjacent groundwater because of these additional chemical compounds (Becerra-Castro, 2015; Christou et al., 2017; Desbiolles et al., 2018; WHO, 2006). On the last decades, contamination from pharmaceutical residues is noticed all over the world and therefore considered as emerging environmental pollution (Bagheri et al., 2016; Ortiz de Garcia et al., 2013; Strauch, 2011). Although pharmaceutical residues found in many aqueous environments worldwide, Israel, as a leading leading country in allocating treated wastewater for agricultural irrigation, which is in essence as a "living laboratory", and committed to carefully examining the various issues and implications that are tightly bound to unlimited wastewater reuse (Cohen et al., 2016).

In this paper, we examine the complexity of the implications arising **__from recycling wastewater,__** which contains discrete amounts of pharmaceutical residues and their metabolites, that should treated cautiously. To clarify these implications, we will answer the following questions: What other components are present in effluents? At what concentrations? What are their degradation products? How chemically stable, and how toxic are these degradation products? How do these components and degradation products affect the irrigated environment? What are the regulated parameters in water regulation? Do they provide any parameters in the context of recycled water? Do the regulated parameters indeed define high-quality effluents?

**3.1 What components are present in high-grade effluents? At what concentrations?**

The wastewater flowing into wastewater plants carries pharmaceutical residues, includingantibiotics, anti-inflammatory drugs, anti-depressants, sex hormones, lipid regulators and beta-blockers (Desbiolles et al., 2018). Depending on the specific circumstances, studies have shown that about 30–90% of all medications consumed, excreted via the urine and feces, as the parent compound or its metabolites, are introduced to wastewater (Adamczak et al., 2012; Christou et al., 2017).

The conventional treatment procedure for wastewater includes pretreatment, which is filtration and sedimentation, a secondary treatment that is based on biological degradation, and a tertiary treatment that consists of filtration and disinfection. The secondary treatment is the main stage designed to remove dissolved organic compounds. Apparently, it may remove -20% - +80% of the degraded organic compounds, as measured by biological oxygen demand (BOD) (Grandclément et al., 2017). In practice, at the end of the treatment procedure, the treated wastewater is considered "high-quality effluent" as it meets the standards for the main tested parameters, including BOD, COD, TSS, Soluble sodium percentage, Turbidity, N Total, $NO_3^-$, $NH_4^+$, Oil and Grease, etc. (Shakir et al., 2016). Studies have shown that during the biological treatment of wastewater, resistant pharmaceutical, for example ATZ, diazinon (DZN), diclofenac (DCF), carbamazepine (CBZ), metoprolol (METOP), do not degrade easily (Grandclément et al., 2017) and practically, different pharmaceutical residues have been detected in the aquatic environment around the world (Comber et al., 2018; Yang et al., 2017). The concentration of pharmaceutical residues in wastewater range from 100 to 100000 ng/L or even microgram/L in treated wastewater (Desbiolles et al., 2018; Lamm et al., 2009; Shafrir and Avisar, 2012; Souza et al., 2018; Zhang et al., 2008).

**3.2 Degradation products – Are they chemically stable? Are they toxic?**

Despite their chemical stability, some of the pharmaceutical residues might be degraded while flowing with the wastewater (Kosjak & Heath, 2011; Wang and Wang, 2016). In addition, natural degradation can caused by solar radiation, wastewater acidity (pH), or integration of additional organic compounds such as humic acid (Gozlan et al., 2010, 2014). Variation in wastewater composition influences degradation processes differently, although the parent compound will degrade preferentially to different specific degradation products under different conditions. This further complicates the monitoring and analysis of degradation products (Gozlan et al., 2013, 2016). The following example with the degradation of Amoxicillin (AMX) is emphasize part of the complications. AMX is the active ingredient in commonly consumed antibiotics worldwide. Surprisingly, it has never been detected in wastewater or effluent. It was found that AMX degrades easily due to opening of its strained β-lactam ring during its hydrolysis. As a result, its chemical identity changes, and its degradation product diketopiperazine-2', 5' (ADP), which is neither stable nor toxic, is produced (Gozlan et al., 2013, 2016; Lamm et al., 2009). This product continues to degrade into a few unstable and nontoxic degradation products, and one stable and toxic product, ADP3 (Gozlan et al., 2016; Lamm et al., 2009).

ADP3 is the major degradation product of AMX and is defined as a stable, biologically active compound. It has been detected in surface water as well as in groundwater and represents its parent molecule's distribution. The case of AMX exemplifies the essence of degradation processes that might affect the parent compound and its degradation products while in the environment. It emphasizes the difficulty in detecting compounds and their degradation products in their stable form, which might lead to the development of resistant bacteria and even cause other possible health hazards to human, wild and domestic animals (Gozlan et al., 2010).

As already noted, there are many types of degradation products that might be more chemically stable than their parent compound (Gozlan et al., 2013). This stability is alarming since such products might form a chemically active structure that is toxic and threatens human and animal health. Another alarming outcome of these degradation products' chemical stability might be the development of resistant bacteria when the parent product is an antibiotic (Gozlan et al., 2010; Lamm et al., 2009). Active degradation products might form during treatment by advanced oxidation processes (AOPs), although this is the suggested treatment type for the degradation of drug residues. Therefore, learning and investigating degradation processes under a variety of conditions, in addition to developing monitoring and measurement methods to detect them, is of great importance (Gozlan et al., 2010).

Effluent that has been allocated for agricultural irrigation might be rich in chemically stable drug residues and their degradation products. These compounds then flow and drain into the environment, infiltrating with the water and mixing with surface and groundwater, especially next to cultivated areas (Avisar et al., 2009; Zhang et al., 2008). Measuring and detecting all degradation products in the environment is difficult since many have not yet been identified (by name or by chemical definition). Our knowledge is restricted and there are literally hundreds of unknown compounds in the aquatic environment that cannot be detected or measured (Lamm et al., 2009; Yin et al., 2017), and therefore cannot know whether these unknown compounds are toxic or how their presence in the environment influences human health and the environment (Grossman and Rurman, 2007). On the other hand, we do know that pharmaceutical residues in the influent flowing into wastewater treatment plants may exert toxic or inhibitory effects on activated sludge bacteria (Yang et al., 2017). There are also studies that showed that long-term exposure to very-low concentration of pharmaceutical residues in the aquatic environment pose risk on the aquatic ecosystem (Strauch, 2011), and mixture of pharmaceutical residues and accumulation in soils, can threaten the environmental and human health (Beccera-Castro et al., 2015). Awareness to these dissolved and suspended organic compounds and their degradation products in water, and their stable structure which is not mineralized to water and $CO_2$ during dedicated treatment (secondary, tertiary or AOPs), emphasize the importance of exploring the toxicity of these products in water.

**3.3 How do residual drugs and their degradation products affect the irrigated environment?**

[revised manuscript text omitted]

5 In most of the world, pharmaceutical residues and their degradation products found in treated wastewater and has no regulation restricting them (Blair et al., 2015; Grandclément et al., 2017; Yang et al., 2017). It is particularly because of their low concentration, their large variety of chemical structures and the unknown degradation products, that posing a challenge on the analytical methods and though leave them out of regulation (Beccera-Castro et al., 2015; Blair et al., 2015). On the last decades, with analytical development and increasingly frequent medical prescriptions, and medication consumption, many studies

10 detected pharmaceutical residues and their metabolite, in contaminating soils (Blair et al., 2015), agricultural products (Becerra-Castro et al., 2015; Christou et al., 2017) and water resources (Ortiz de Garcia et al., 2013; Strauch 2011). It is already clear that some pharmaceutical residues cause health implications and hazards and the research must deepen the understanding of this context. Pharmaceutical residues mixing can synergistically enhance their toxicity, similarly to the phenomena happened with patients who taken mix of medications, although the residues are on order of magnitude lower than

15 administrated pharmaceutical (Comber et al., 2018; Ebele et al., 2017). It is also clear that the possible implications need to be consider for future regulation although there is much more we do not know than what we know (Desbiolles et al., 2018). The consideration of pharmaceutical residues in wastewater effluent got expression recently, while some pharmaceutical residues start to be monitored by European counties in the field of water policies. In the national level of Switzerland and England, monitoring and treatment efforts to remove or dilute antibiotics from the treatment-plant effluent, has already held (Comber et

20 al., 2018; Swiss Federal Concil, 2016).

**4. Where do we go from here?**

It is obvious that in the coming years, additional regulatory requirements will be needed to monitor and treat pharmaceutical residues. The required treatment will be applied at the treatment plants, or at the wastewater source in the case of industries or

25 hospitals (Gerrity and Snyder, 2011). Treatment at the wastewater source will need to be performed on smaller volumes and with specific contaminants to degrade specific pharmaceutical, to make the process more efficient and less expensive.

The main difficulty in degrading pharmaceutical is their physico-chemical properties. As mention earlier, these compounds are only partially degraded by the common biological degradation process  and the effluent leaving the plant designated for agricultural irrigation with no restrictions, is actually contains a mixture of pharmaceutical residues. Many studies promoting

30 researches to prevent or eliminate pharmaceutical residues to get to the effluent water. In the last few years, research groups worldwide have been developing combined methods to degrade and remove pharmaceutical residues at the treatment plant as well as at the wastewater sources (industry and hospitals. Biological treatments such as activated sludge, membrane bioreactor

treatment (MBR) are examples of biological degradation in treatment plants, in which operating conditions effect removal efficiency (Ejhed et al., 2018; Grandclément et al., 2017). Membrane filtration such as reverse osmosis reactor is also one of the solution under research (Cerro-Lopez et al., 2019). On the nanomaterials techniques there are metal nanoparticles, carbon nanotubes and nano filters which all of them found to be efficient removal for some of the contaminants (Bagheri et al., 2016).

5  One of the solutions stem from examinations of different technological aspects, focus on the development and improvement of Advanced Oxidation Process (AOPs). Some of AOPs techniques, can effectively treat the problem and degrade pharmaceutical residues, although not all of them (Lester et al., 2013; Siedlecka et al., 2018), Anyway, it comes up from different studies, that almost all of the treatment possibilities describe above, are often not sufficient ensure high removal of the pharmaceutical and therefore much of the technique improves by combining two or more treatment processes to one hybrid

10  treatment procedure (Grandclemente et al., 2017; Zucker et al., 2015).

**5. Awareness**

The most important task from the above manuscript is to raise awareness of the possible risks folds in using secondary effluent for agriculture irrigation. Arguments that pharmaceutical residues concentrations are too small to influence human health are

15  easy to make, but have never been proven. There have not yet been enough studies to confirm or reject this argument. It is therefore important to emphasize that even though pharmaceutical residues and their degradation products, are not supposed to be in the drinking water at all, they do exist there, as a mixture. However, awareness to the possible danger impose by this unknown mixture must be carefully consider, before it can be ignored.. The authors think that "conducting business as usual", i.e., persisting with a regulatory attitude that ignores OMPs in effluent, is tantamount to "turning a blind eye" and allowing the

20  overwhelming neglect to worsen.

Globally, renewed legislation and the establishment of concentration permits for pharmaceutical residues contaminants in water resources and effluent are essential (Gozlan et al., 2014). Preparation for this essential stage includes comprehensive studies of the different chemical groups of the widespread drugs and their degradation products, method development to identify and measure them, awareness of the analytical restrictions, and testing the possible influence of different

25  pharmaceutical residues mixtures on human health. Similar source treatment of hospital and industrial wastewater is a must, and obliging the drug industry to treat their waste and establish specific permits for pharmaceutical residues concentrations in wastewater are critical to successfully implementing treatment methods to remove it. This will enable the production of high-quality effluent that is readily available for any use.